# Meta-Analysis of Steel Fiber-Reinforced Concrete Mixtures Leads to Practical Mix Design Methodology

**DOI:** 10.3390/ma14143900

**Published:** 2021-07-13

**Authors:** Emilio Garcia-Taengua, Mehdi Bakhshi, Liberato Ferrara

**Affiliations:** 1School of Civil Engineering, University of Leeds, Leeds LS2 9JT, UK; 2AECOM, Los Angeles, CA 90071, USA; Mehdi.Bakhshi@aecom.com; 3Department of Civil and Environmental Engineering, Politecnico di Milano, 20133 Milan, Italy; liberato.ferrara@polimi.it

**Keywords:** artificial intelligence, concrete, database, FRC, mix design, fibers, proportioning

## Abstract

The analysis of hundreds of SFRC mixtures compiled from papers published over the last 20 years is reported. This paper is focused on the relationships between the size and dosage of steel fibers and the relative amounts of the constituents of SFRC mixtures. Multiple linear regression is applied to the statistical modeling of such relationships, leading to four equations that show considerable accuracy and robustness in estimating SFRC mixture proportions as a function of fiber content and dimensions, maximum aggregate size, and water-to-cement ratio. The main trends described by these equations are discussed in detail. The importance of the interactions between aggregates, supplementary cementitious materials, and fibers in proportioning SFRC mixtures, as well as implications for workability and stability, are emphasized. The simplicity of these data-driven equations makes them a valuable tool to guide the proportioning of SFRC mixtures. Their predictive performance when used together as a data-driven mix design methodology is confirmed using a validation dataset.

## 1. Introduction

About sixty years ago, the first pioneer studies presenting a rational approach to the concept and applications of Fiber Reinforced Concrete (FRC), and steel FRC (SFRC) in particular, based on fracture mechanics were published [1,2,3,4,5]. Since then, consensus has been reached at an international level on a performance-based methodology for structural design, which has been incorporated into modern worldwide accepted codes and guidelines, such as the Model Code and the ACI 544 documents [6,7,8,9,10,11]. The proportioning of SFRC mixtures has nevertheless remained a sort of grey area so far, which has not received likewise attention.

Among concrete practitioners, the dominant way of thinking in terms of proportioning SFRC mixtures remains based on one single parameter (the dosage of fibers) which dictates the design of the mixtures. Also, the misperception that a good SFRC mixture is basically the outcome of fiber addition to a good parent plain mix, spiced up with some admixture adjustment to comply with fresh state performance, still has significant prevalence. In order to get over these two misperceptions, a methodology for the proportioning of SFRC mixtures is needed which links the main mix design variables to the fiber dosage and fiber dimensions and these, in turn, to the most significant indicators of the FRC performance in the fresh and hardened state. However, the first and necessary step towards a performance-based proportioning method is surely the in-depth analysis and modeling of the relationships between the relative proportions of the different SFRC constituents, which is the focus of this paper. Only then can these variables be connected to performance specifications as part of a conceptually robust framework.

In this framework, it is worth remarking that the fresh state performance of SFRC mixtures needs to be regarded as equally important to the hardened state properties. First of all, it has to be considered within the broader category of modern SFRCs, which includes materials with enhanced performance in the fresh state, such as self-compacting FRC and Ultra High Performance (Fiber Reinforced) Concretes (UHPFRCCs) [12,13]. In this context, the achievement of adequate levels of fresh state performance, which is inextricably related to the mechanical properties and structural performance, is critical to the design of the mixture. In fact, an adequate fresh state behavior, is in most cases, determining to the obtainment of the desired mechanical behavior. A tailored and rational mix design methodology for SFRC can allow the synergies between fresh and hardened state performance to be exploited at their best, e.g., through the possibility of aligning, thanks to a suitably balanced material rheology, the fibers along the casting flow direction, from which an optimal structural performance of the material follows [14,15]. Indeed, a good mix design methodology must be deeply rooted in the current practice of concrete mix design and batching technologies, and that is the reason why a systematic analysis of all the accumulated experience of designing, trialing, adjusting, and producing SFRC mixtures, which is implicit in the vast amount of mix designs used in different studies over the years, has an enormous value.

In order to address these issues, the project “Optimization of Fiber-Reinforced Concrete using Data Mining” is underway with funding received from the Concrete Research Council through the American Concrete Institute Foundation [16]. The study reported in this paper has been conducted as part of this project. It has at its core the application of data mining and statistical modeling techniques to analyze and model the relationships between mix design variables, fresh state performance, and mechanical properties, following the methodological principles applied to self-consolidating concrete in previous work [17,18].

The first step towards this quite challenging, holistic task consisted in building up a database of SFRC mixture proportionings and their most relevant fresh and hardened state properties from the papers published in the last twenty years, this period marking the maturity of the SFRC technology and of its applications. A meta-analysis was then performed, aimed not only at extracting the most significant characteristics of the SFRC mixtures successfully produced, studied, and employed so far, but also to find the correlations among the different mix design variables. Such an analysis was conceived as a two-stage process. As mentioned above, the focus of this paper is on the relationships between the size and dosage of steel fibers and the relative amounts of the other constituents of SFRC mixtures. This work was intended as prodromal to a further analysis and exploitation of the database and of the meta-analysis process to derive the correlations between the most significant mix design parameters, identified and cross-analyzed as hereafter, to the indicators of performance of SFRC mixes in the fresh and hardened state to establish a performance-based mix design methodology which needs to be incorporated into structural design approaches for both design predictions and quality control tools. This second stage of the study will be addressed in a separate paper.

## 2. Methodology

The methodological principles applied to this study were those of a meta-analysis, that is, an analysis of experimental results collected from multiple different studies and aggregated in a larger set of results or database [19]. The database compiled as part of this study incorporated the variability that arises between mixtures designed following different approaches and considering materials from different sources, which made it widely representative of SFRC mixtures. This database was preprocessed and cleaned in order to detect and discard any anomalous cases or misreported values. After that, statistical techniques were applied to analyze the information in the database by modelling the relationships between different mixture proportioning parameters.

### 2.1. Data Collection and Structure of the Database

A database of SFRC mixtures was compiled from papers published between the years 2000 and 2019. The main sources used to search for papers were http://www.sciencedirect.com (accessed on 2 September 2019) and the index of papers published by the American Concrete Institute (ACI) journals, with the search terms being “steel fiber reinforced concrete” and its alternative spellings, “SFRC” and “steel FRC”. The complete list of sources from which the database was compiled is given in the Appendix A that accompanies this paper.

The main challenges were the selection of papers (so that sufficient information to be collected was available and was representative of a wide range of SFRCs) and the treatment of incomplete information [20]. The size of the database (number of cases) was also carefully considered. Datasets comprising less than 30 cases are generally regarded as insufficient, however, it is widely accepted that excessively large case numbers can be problematic and make statistical inference less robust [21]. As a consequence, it was decided that the database to be compiled for this study had to include no less than 30 cases and no more than 1000 cases.

The SFRC mixtures database was structured so that each row (or case) corresponded to a different SFRC mixture, while the columns (variables) corresponded to the parameters describing the mixture proportions and other characteristics describing each mixture, including the dimensions of the fibers.

### 2.2. Definition of Variables

The variables related to the proportioning of SFRC mixtures were expressed in terms of relative weights per unit volume of concrete, in SI units (e.g., kg/m^3^). These included the contents of: cement, water, ground granulated blast furnace slag (GGBS), silica fume, fly ash, limestone powder or filler, fine aggregate (sand), coarse aggregate (gravel), superplasticizer, air entraining admixture, retarder, and viscosity modifying admixtures (VMAs). Fiber contents were expressed as the volume fraction of fibers, in percentage (*V_f_*).

The sum of cement, GGBS, silica fume, fly ash, and limestone powder contents was stored in a variable called ‘binder’, and the difference between the binder content and the cement content was stored in a variable called ‘SCMs’, which stands for ‘supplementary cementitious materials’ [22].

The maximum aggregate size was expressed in millimeters (mm), and the size of the fibers used in each SFRC mixture was described by the following variables: fiber length and diameter in millimeters (mm), aspect ratio (non-dimensional), and fiber shape (using broadly descriptive terms such as hooked, or straight).

The type of cement was stored in a categorical variable, codified as CEM I, II, III, or IV in correspondence with the main categories in European standards [23].

### 2.3. Cleansing of Data and Descriptive Analysis

A preliminary analysis of the database was carried out to detect and discard cases where information was clearly flawed or not consistent with the vast majority of mixtures (e.g., misreported values).

The presence of missing data was inevitable because retrieving the values for all variables was not always possible. Cases where most of the information was missing were discarded. Those cases with a small part of the information missing were treated by means of multiple imputation [24,25], a process by which the missing values were replaced by estimates based on the exploitation of correlation with other variables, known relationships between them or the underlying phenomena they represent, or a combination of both.

A descriptive analysis of the database, variable by variable, was carried out to determine typical ranges and to identify any outliers. Histograms and cumulative frequency distribution plots were used to investigate the frequency distribution of each variable. For each variable, the median was calculated as indicative of its central value, and the percentiles corresponding to cumulative frequencies of 5% and 95% were calculated to define the representative range of variation for each variable, rather than absolute minima and maxima. Potential outliers were identified based on the so-called Tukey’s fences, also known as the interquartile range method for outlier detection [26,27,28].

### 2.4. Statistical Modeling and Contour Plots

The statistical modeling and subsequent analysis reported in this paper are concerned with the relationships that exist between different variables describing SFRC mixtures in terms of the relative amounts of their constituents and the characteristics of such constituents.

Multiple linear regression was used for modeling such relationships by fitting descriptive equations to the values in the database, with two equally important objectives. First, to produce plots which could be used to visualize, describe, and interpret those relationships. Second, to propose a consistent set of equations which can be used as part of the SFRC mix proportioning process. The same modeling process was followed in all instances. Initially, the regression models included all pairwise interactions between variables and their quadratic terms. Stepwise regression was then used to simplify the initial models, sequentially removing those terms and interactions that were not statistically significant [21]. The final models correspond to the equations reported in this paper, and were described and discussed using the corresponding contour plots.

## 3. Preparation and Descriptive Analysis of the Database

The database of SFRC mixtures compiled in this study, after the identification and removal of anomalous or severely incomplete cases, comprised 766 different cases extracted from more than 100 papers published between 2000 and 2019.

### 3.1. Imputation of Missing Values

The prevalence of missing data was low: 56.6% of the cases in the database were complete and, overall, only 7.8% of the information was missing. The percentages of missing data for different variables are shown in Table 1, which shows that missing values were mostly concentrated on two variables: cement type and maximum aggregate size.

Missing data regarding the cement type could not be remedied, due to the relatively high percentage of data that was missing and the categorical, non-quantitative nature of the variable.

The imputation of missing maximum aggregate size values was possible in the majority of cases. Some of these cases corresponded to SFRC mixtures which contained no coarse aggregate, and therefore a maximum aggregate size of 4 mm was imputed. However, a few of these cases had no coarse aggregate because they were ultra-high performance mixtures, and the maximum aggregate size was set to 1 mm instead, as this is generally the case for UHPFRC—the only aggregate they contain is fine sand [13,29]. On the other hand, in those cases corresponding to SFRC mixtures with both fine and coarse aggregates, the imputation of missing maximum aggregate size values was done by using a regression equation fitted to the subset of complete cases.

The imputation of fine and coarse aggregate content in the few cases where both were missing (3.2%) was done using other information reported in the papers from which they were extracted. The total volume of aggregates per unit volume of SFRC was determined by subtraction, and the relative proportions of aggregates were estimated by approximating the combined grading of solids to the Bolomey’s curve [30] with a parameter of 12, considering an average specific weight of 2.7 for both fine and coarse aggregates.

### 3.2. Descriptive Analysis

Table 2 presents a summary of the information in the database. Five values are given as descriptors of the most significant variables and parameters defining the composition and characteristics of the SFRC mixtures in the database.

The median is the value below which 50% of the cases fall, thus representing the middle value of the different parameters. The minimum and maximum limits of the so-called ‘representative range’ correspond to the 5th and 95th percentiles, i.e., the values between which 90% of the cases are found. On the other hand, the ‘typical range’ is given as indicative of the most common values for each parameter. The minimum and maximum limits of the ‘typical range’ correspond to the 25th and 75th percentiles respectively, therefore comprising 50% of the cases.

In almost 42% of the cases, the Portland cement type was not reported. However, in 94% of the cases for which this information was available, it was either CEM I or CEM II. Besides, CEM I, in particular, was the most prevalent, being 38% more frequent than CEM II among the SFRC mixtures in the database. Regarding the composition of the binder, it is also interesting to note that 52.7% of the cases corresponded to mixtures which incorporated no SCMs in addition to cement. Representative values for the SCMs content in Table 2 were determined based on the subset of cases which included SCMs.

Superplasticizer was used in 79% of the cases, confirming that the use of this family of chemical admixtures has become common practice in the proportioning of SFRC mixtures. As the values in Table 2 show, the frequency distribution of superplasticizer contents was strongly skewed, as small values were far more common than high values. This was attributed to the fact that favoring low superplasticizer contents to control workability is advantageous for different reasons: stability issues when dosed in excess (which is particularly critical in the case of self-compacting FRCs) [17,31,32], the relatively high cost of admixtures, and diminishing returns beyond the saturation point [33,34,35]. In fact, taking the median values, the ratio of superplasticizer to cement content was 1.125%, which is consistent with the typical saturation point dosages, normally between 1–1.5% [35,36].

Regarding the steel fiber content, the distribution of *V_f_* values in the database was also skewed, since lower values were more common than high values: the median (0.51%) was closer to the minimum (0.25%) than it was to the maximum (2%) of the representative range. In 50% of the cases, the fiber volume fraction was not higher than 0.51%. In 75% of the cases, *V_f_* was lower than 1.0%, and it was lower than 1.6% in 90% of the cases.

## 4. Statistical Models, Interpretation and Discussion

The process of statistical modeling is not straightforward when it is concerned with a multivariate dataset, as was the case in this study. Multiple linear regression could be applied to obtain as many regression equations as there are variables in the database, and even more if different subsets of other variables were considered as predictors or covariates. A high number of alternative models were explored to find refined models that satisfied three conditions: (i) being satisfactory in terms of goodness of fit, that is, proving reasonably accurate in fitting the data; (ii) not presenting multicollinearity problems [37]; (iii) presenting simple specification, being relatively simple and not including many terms and interactions.

After various attempts at preliminary analyses, it turned out that three models in particular satisfied the above conditions, and also constituted a structured, coherent set of equations that could be used to produce reasonable SFRC mix proportioning estimates without being excessively complex. The following sections present and discuss these models.

### 4.1. Coarse-to-Fine Aggregate Ratio

The coarse-to-fine aggregate ratio, or gravel/sand (*G*/*S*) ratio, was considered a relevant parameter to investigate, as it is generally representative of the combined aggregates grading, which contributes to the cohesiveness of fresh SFRC mixtures and, in turn, affects their workability and stability in the fresh state [38]. Initially, an exploratory analysis looking into the relationship between the *G*/*S* ratio, the fiber dimensions, and their dosage was carried out. However, it was concluded that there were no statistically significant associations between the *G*/*S* ratio and these variables. The only two variables which turned out to be clearly correlated with the *G*/*S* ratio were the maximum aggregate size and the coarse aggregate content. The scatterplot in Figure 1 shows the values of the *G*/*S* ratio versus coarse aggregate content for the SFRC mixtures in the database. A clear trend is observed, showing a direct, positive correlation between the two parameters.

A multiple regression analysis was carried out to obtain an expression relating the *G*/*S* ratio to the coarse aggregate content and the maximum aggregate size. The following equation was obtained (R-squared = 0.92), with an error term of ±0.10 for the *G*/*S* estimate:(1)$$\frac{G}{S}=\left(20.3+22.27\;m+0.269\;G+0.001\;G^{2}-0.029\;m\;G\right)\times 10^{-3}\pm 0.10$$
where *G* and *S* are the coarse aggregate (gravel) and fine aggregate (sand) contents, respectively, expressed in kg/m^3^, and *m* is the maximum aggregate size expressed in mm.

Figure 2 shows the response surface obtained by plotting Equation (1) against the coarse aggregate content and maximum aggregate size. Assuming the maximum aggregate size at 14 mm, an increase of 100 kg/m^3^ in the coarse aggregate content corresponds to an average increase in the *G*/*S* ratio between 0.12 and 0.19, becoming more pronounced the higher the coarse aggregate content. However, the relationship between *G*/*S* ratio and coarse aggregate content was also slightly affected by the maximum aggregate size, as can be observed in Figure 2. For example, increasing the coarse aggregate content from 800 to 900 kg/m^3^ was correlated with the *G*/*S* ratio increasing by 0.17 for a maximum aggregate size of 10 mm, whilst this increase was 0.14 for a maximum aggregate size of 20 mm.

### 4.2. Coarse Aggregate Content

After modeling the *G*/*S* ratio, an investigation into the relative amounts of aggregate was necessary, as this determines the relative volume of aggregates and, in turn, the relative volume of paste, which is crucial to fresh and hardened state performance [39]. Furthermore, the difference between the shape and dimensions of aggregates and fibers justified the interest of quantifying the effect of fibers on the aggregates content and grading [40,41]. Preliminary analyses were conducted to model the sand, gravel, and total aggregate contents as a function of the other mixture parameters. The most satisfactory model in terms of goodness of fit and simple specification was obtained for the coarse aggregate content.

The multiple regression analysis to model the gravel content produced an equation which relates this variable to the maximum aggregate size, the fiber dimensions, and the fiber volume fraction, all of which were statistically significant. After considering all quadratic terms and interactions, and removing all non-significant terms, the following equation (R-squared = 0.76), with an error term of ±129 kg/m^3^ for the estimate, was obtained:(2)$$G=-95-2.21m^{2}+V_{f}\left(4.15l_{f}-1.35\lambda_{f}-6.44m\right)+m\left(120-0.15l_{f}-0.22\lambda_{f}\right)\pm 129$$
where *m* is the maximum aggregate size (mm), *V_f_* is the fiber volume fraction (percentage), *l_f_* is the fiber length (mm), and *λ_f_* is the fiber aspect ratio.

#### 4.2.1. Fiber Dimensions and Gravel Content

Figure 3 shows the contour plots for the gravel content versus fiber length and aspect ratio, obtained by plotting Equation (2) considering four different values for the fiber volume fraction: 0.25%, 0.5%, 1.0%, and 1.5%, and assuming a maximum aggregate size equal to the median of the values in the database, i.e., 14 mm.

It can be observed that the relative effect of fiber length and aspect ratio on the gravel content changes with the fiber volume fraction. For low to medium fiber dosages (Figure 3a,b), the determining factor was the aspect ratio: lower aspect ratios were associated with higher gravel contents. For medium to high fiber dosages (Figure 3c,d), both the aspect ratio and the fiber length were associated with changes in the gravel content, and longer fibers were clearly associated with higher gravel contents.

Increasing the fiber aspect ratio was found to be associated with decreasing the gravel content, which implied increasing the sand content and the relative volume of paste, which generally makes for more cohesive mixtures. Therefore, fibers with higher aspect ratios were found to generally require the SFRC mixture to be more cohesive, which was linked to workability and fresh state stability requirements [42,43]. A similar rationale is applicable to the relationship between the fiber length and the gravel content. The use of shorter fibers, particularly when the fiber dosage is not small, implies that the number of fibers per unit volume is high, hence requiring the mixture to be more cohesive, proportioned with more sand and paste, and less gravel. Such an interpretation is also consistent with previous findings regarding the sensitivity of the voids ratio to the fiber dimensions and content [44].

In the discussion above, references have been made to low and high fiber dosages in relation to Figure 3. Equation (2) can be used to quantify the fiber volume fraction above which the effect of fiber length on gravel content becomes more pronounced. By differentiating the gravel content, as per Equation (2), with respect to fiber length and equating to zero, such threshold value for the fiber volume fraction was obtained, which depends on the maximum aggregate size (*m*):(3)$$\frac{\partial G}{\partial l_{f}}=0\;\;;\;\;4.153V_{f}-0.148m=0\to V_{f}=\{\begin{array}{c} 0.36\%\;\;if\;m=10\;\mathrm{mm}\\ 0.50\%\;\;if\;m=14\;\mathrm{mm}\\ 0.71\%\;\;if\;m=20\;\mathrm{mm} \end{array}$$

#### 4.2.2. Fiber Volume Fraction and Gravel Content

Figure 4 shows the contour plot obtained by plotting gravel contents, as per Equation (2), with respect to the fiber volume fraction and the fiber length, assuming the median maximum aggregate size and the median fiber aspect ratio values in Table 2. Three dashed lines have been added as reference, to facilitate the interpretation. The horizontal dashed line corresponds to a fiber volume fraction of 0.54%, as per Equation (3), below which the gravel content was not significantly sensitive to the fiber length and was mostly in the range of 800–850 kg/m^3^. Gravel contents within this range were also associated with higher fiber volume fractions, as long as the fiber length was in the region of 40 mm.

Figure 4 also shows that shorter fibers in moderate to high dosages were generally associated with markedly diminishing gravel contents. This indicated that SFRC mixtures with moderate to high contents of shorter fibers are generally proportioned to have less gravel and higher relative volumes of sand and paste, which was attributable to increased cohesiveness requirements. On the other hand, moderate to high dosages of longer fibers were associated with an increased gravel content, typically ranging from 850 to 950 kg/m^3^, as shown in Figure 4. This can be attributed to the size compatibility of 40–60 mm long steel fibers with the average particle size of the combined aggregates grading in SFRC mixtures.

#### 4.2.3. Maximum Aggregate Size and Gravel Content

Figure 5 shows the main effects plot for the gravel content with respect to the maximum aggregate size, which represents the mean gravel content as per Equation (2). The red line corresponds to the average estimates, whilst the grey band represents the margin of error of Equation (2). This plot was useful to discuss the general trend in the relationship between the gravel content and the maximum aggregate size disaggregated from the effect of those terms in Equation (2) that do not include the maximum aggregate size.

As shown in Figure 5, increasing the maximum aggregate size was associated with increasing gravel contents, however this trend was not linear. For maximum aggregate sizes above 14 mm, the curvature of this relationship would be compatible with the gravel content not being affected by changes in the maximum aggregate size, given the margin of error. In consequence, it can be said that there is a direct, positive correlation between the gravel content and the maximum aggregate size in SFRC mixtures with a maximum aggregate size not higher than 14 mm.

### 4.3. Binder Content

Following the models concerned with the aggregates, the next step was the modeling of the relative amounts of binder. Initial modeling efforts included the cement content, SCMs content, the water-to-binder and water-to-cement ratios as a function of the maximum aggregate size, the aggregates content, the fiber dimensions, and the fiber volume fraction. The parameter which led to the best model in terms of accuracy and simplicity was the total binder content.

Multiple linear regression analysis was applied to the binder content (expressed in kg/m^3^) in relation to the abovementioned variables, initially considering all quadratic and interaction terms. After simplifying the model by stepwise regression, the following equation was obtained (R-squared = 0.90), with an error term of ±56 kg/m^3^ for the estimate:(4)$$B=262.1-6.11m+\left(19800S-10.51S^{2}-4328G+3.66G^{2}-7.45\;S\times G\right)\times 10^{-4}\pm 56$$
where *B* is the binder content (expressed in kg/m^3^), *m* is the maximum aggregate size (mm), and *S* and *G* are the sand and gravel contents (expressed in kg/m^3^), respectively.

It is worth mentioning that the variables related to the fibers (volume fraction, fiber length, and fiber aspect ratio) do not appear in Equation (4) because they were found not to have a statistically significant effect on the binder content. However, this does not mean that these variables do not affect the binder content, because their effect in Equation (4) is implicit through their effect on aggregates contents (see Equation (2)).

#### 4.3.1. Maximum Aggregate Size and Binder Content

Increasing the maximum aggregate size was found to be directly associated with moderate reductions in the binder content. On average, as per Equation (4), increasing the maximum aggregate size from 10 to 20 mm was associated with the binder content decreasing by 61 kg/m^3^. This represented an average decrease in the binder content by 13.6% when the median binder content of 450 kg/m^3^ was taken as reference.

#### 4.3.2. Aggregates and Binder Content

Equation (4) was used to produce contour plots relating the binder content to the fine and coarse aggregate contents, which are shown in Figure 6. The same range was considered for both fine and coarse aggregate contents: between 300 and 1200 kg/m^3^. However, not all combinations of sand and gravel contents in these ranges were realistic, as it is not possible to have 1200 kg/m^3^ of sand and 1200 kg/m^3^ of gravel at the same time in a normal weight SFRC mixture. Therefore, the interpretation of these contour plots had to be restricted to a feasible region. This was done by superimposing the following condition: the sum of the relative amounts of all constituents had to be consistent with the average specific weight of the SFRC mixtures in the database, which was 2470 kg/m^3^. Assuming the median values given in Table 2 for the water content, the superplasticizer content, and the fiber volume fraction, and a margin of variation established at ±150 kg/m^3^ for the binder content, the condition to be imposed was as follows (*B*, *S*, and *G* expressed in kg/m^3^):(5)2096.5≤B+S+G≤2396.5

The plots shown in Figure 6 assume two different values for the maximum aggregate size of 10 and 20 mm, and the greyed-out areas correspond to sand-gravel-binder combinations that are not compatible with the condition given in Equation (5). Both being exactly the same shape, the only difference is that contours for the maximum aggregate size of 10 mm are stepped down a bracket when compared to the 20 mm case. In fact, it is stepped down by 61 kg/m^3^.

In both cases, higher binder contents were associated with gravel contents being lower than sand contents. This means that the coarse-to-fine aggregate ratio tended to decrease with increasing binder contents. On the other hand, lower binder contents were associated with mixtures in which the sand and gravel contents were not too dissimilar, with the coarse-to-fine aggregate ratio being in the range of 0.7–1.3. This can be explained in the following terms: when the binder content is low, it is necessary for the mix of all aggregates to have a better grading for ensuring an adequate cohesiveness of the mixture, thus moving in the direction of gravel-to-sand ratios closer to unity. It is also interesting to note that cases in which the binder content is low correspond to more conventional, ordinary SFRCs, as opposed to higher binder contents which are often associated with mixtures with higher performance requirements.

### 4.4. Superplasticizer Content

Superplasticizer content values in the database were highly scattered. In terms of the representative range (Table 2), the maximum value was more than ten times the minimum. This was due to the fact that different superplasticizers can be very different in terms of their recommended dosage ranges and optimal dosages. Nevertheless, the multiple regression analysis for the relative amount of superplasticizer (*SP*, kg/m^3^) led to a reasonably accurate equation (R-squared = 0.77), with an error term of ±3.7 kg/m^3^:(6)$$SP=\left(3930+27.53\;B+0.032\;B^{2}+0.67\;W^{2}-0.42\;B\times W+0.38\;B\times \lambda_{f}-1.50\;W\times l_{f}-0.68\;W\times \lambda_{f}+17.23\;m\times l_{f}-10.72\;m\times \lambda_{f}\right)\times 10^{-3}\pm 3.7$$
where *SP*, *B*, and *W* are the relative contents of superplasticizer, total binder, and water, respectively (all expressed in kg/m^3^), *m* is the maximum aggregate size (mm), *l_f_* is the fiber length (mm), and *λ_f_* is the fiber aspect ratio.

Equation (6) was used to produce the contour plots shown in Figure 7, which represent the estimated superplasticizer content in kg/m^3^ as a function of the water and total binder content. For illustration, two different values for the maximum aggregate size (12 mm and 20 mm), and two fiber sizes (65/60 and 80/30, the first number being the aspect ratio, the second being the length in mm), were considered.

The greyed-out areas correspond to unusual water and binder content combinations, based on the representative ranges in Table 2: water contents out of the 140–237 kg/m^3^ range and water-to-binder ratios out of the 0.2–0.6 range. Also, two particularly relevant contour lines are highlighted. The dashed white line represents a superplasticizer content of 2.5 kg/m^3^, which, to the right-hand side, defines a region corresponding to cases generally associated with low or no superplasticizer contents (colored in dark blue). Since the margin of error for the estimation of superplasticizer content as per Equation (6) was 3.7 kg/m^3^, estimates of 2.5 kg/m^3^ or below were not necessarily different from zero. On the other hand, the dashed black line represents an estimated superplasticizer content of 10 kg/m^3^, which, to the left-hand side, defines a region corresponding to cases associated with high superplasticizer content (colored in green in Figure 7).

Combinations of a high binder content and a low water content were, unsurprisingly, associated with a high superplasticizer content. More interestingly, for any given binder content, Equation (6) can be used to determine the minimum water-to-binder ratio leading to low or no superplasticizer requirements, simply by imposing an estimated superplasticizer content of 2.5 kg/m^3^. Similarly, the water-to-binder ratio below which a high dosage of superplasticizer would be expected can be determined by imposing a superplasticizer content of 10 kg/m^3^ into Equation (6).

SFRC mixtures with 65/60 fibers and a maximum aggregate size of 20 mm were found to generally require the superplasticizer in low to moderate dosages when the binder content is not higher than 400 kg/m^3^ (Figure 7, lower left). For such mixtures, the model estimated that the no-superplasticizer scenario was not an option unless the water content was on the higher end of the spectrum (205 kg/m^3^) and the binder content around 500 kg/m^3^, which corresponds to a water-to-binder ratio of 0.41. Similar observations can be made in relation to the other contour plots in Figure 7. For example, high superplasticizer contents are associated with water-to-binder ratios between 0.21 and 0.24 when the binder content represents 700–800 kg/m^3^ for the cases considered in the figure. In consequence, the model, as per Equation (6), showed that SFRC mixtures where the binder content is 700 kg/m^3^ or higher should be proportioned with water-to-binder ratios not lower than 0.25 in order to avoid excessively high superplasticizer contents. As such high binder content values generally correspond to SFRC mixtures with high performance requirements, this shows that the model works well in estimating the superplasticizer content required for not only the more conventional SFRC mixtures but also the more demanding ones.

## 5. A Data-Driven SFRC Proportioning Method

Once the analysis of the different mix design variables was carried out, a clear methodology for the proportioning of SFRC mixtures became apparent by simply organizing the equations obtained in a logical sequence. This methodology is summarized in Figure 8 in the form of a flow chart.

In the context of proportioning an SFRC mixture to specification, the performance requirements that are specified for the mix will of course inform the inputs to this flow chart, particularly the fiber volume fraction and the water-to-cement ratio. The initial inputs of fiber volume fraction, fiber aspect ratio, and length are inputted into Equation (2) to estimate the gravel content. Then, this gravel content is used along with the maximum aggregate size (*m* in Figure 8) to determine the gravel-to-sand ratio as per Equation (1). Once the sand content is determined, the paste content can be estimated by subtracting the total aggregate content from the total concrete weight per unit volume (for reference, the average value of 2470 kg/m^3^ can be taken). Sand and gravel content, as well as the maximum aggregate size, are then used to determine the binder content as per Equation (4). The binder content can then be subtracted from the paste content to determine the water content. Cement content is then determined by dividing the water content by the water-to-cement ratio desired for the mixture (*w*/*c*). Finally, by subtracting the cement content from the total binder content, the relative amount of supplementary cementitious materials (SCMs) can be determined.

With the objective of validating the SFRC proportioning method proposed in Figure 8 with independent data, a validation dataset of SFRC mixtures was compiled which comprised 100 cases extracted from papers different to those included in the database used to derive the models reported and discussed in this paper. The list of sources from which the validation dataset was compiled is included in the Appendix A that accompanies this paper. Figure 9 shows the comparison between the actual mixture proportions (original values) against the values obtained by using the equations in this paper and the methodology outlined in Figure 8 (calculated values). The dashed lines represent exact equivalence, i.e., original = calculated.

A visual inspection of the scatterplots in Figure 9 shows that the differences between actual mixture proportions and the estimates obtained by using the methodology and equations presented in this paper were relatively small and compatible with the margins of error to be expected. This was confirmed by looking at two parameters that quantify the magnitude of these differences: the root of the mean square error (RMSE) and the average deviation. The RMSE for the gravel content was 153 kg/m^3^, which was comparable to the error term of 129 kg/m^3^ in Equation (2). Similarly, the RMSE for the sand content was 115 kg/m^3^. The average deviation of the estimated gravel and sand contents with respect to their actual values was 15.6% and 12.1%, respectively. The RMSE for the total binder content was 33 kg/m^3^, also consistent with the error term of 56 kg/m^3^ in Equation (4), and the RMSE for the water content was 32 kg/m^3^. The average deviation of the calculated binder and water contents with respect to their original values was 5.9% and 15.2%, respectively. This validation exercise confirmed that the methodology derived from this study was sufficiently robust and reasonably accurate when applied to cases other than those initially considered, which supported its validity.

## 6. Conclusions

For this study, a database of hundreds of SFRC mixtures was compiled from previously published sources. The statistical analysis and modeling herein reported is concerned with the relationships between the SFRC mix design variables, whilst the investigation of correlations between them and performance parameters will be addressed in a separate paper. The main conclusions of this study can be summarized as follows:Steel fibers are typically used in volume fractions between 0.43% and 1.0%, and in 50% of the cases this is lower than 0.51%. In SFRC mixtures, the cement content typically ranges from 360 to 485 kg/m^3^, and SCMs are typically used in contents between 40–100 kg/m^3^. Typical values of the water-to-cement ratio are within the range of 0.36 to 0.50, and in most of the cases, aggregates with a maximum size between 10 and 16 mm are used. Superplasticizers are used in 79% of the cases;Regression models have been obtained for the coarse-to-fine aggregate ratio and relative contents of coarse aggregate, binder, and superplasticizer. These models are robust and effective in producing reasonably accurate estimates (R-squared values between 0.76 and 0.92), and their predictive capacity has been checked against a validation dataset. Given their simplicity, robustness, and good fit to data, they constitute a valuable data-driven methodology to guide the proportioning of SFRC mixtures;The models obtained encapsulate the complex links between fiber dosage, aggregate contents, fiber dimensions, and maximum aggregate size. High fiber aspect ratios, as well as moderate to high dosages of short fibers, were associated with decreasing the gravel content, and increasing the sand content and the relative paste volume, which was linked to workability and fresh state stability requirements;The gravel content of SFRC mixes was found to be practically insensitive to fiber length when the fiber volume fraction was below 0.54%, taking values between 800 and 850 kg/m^3^. Interestingly, gravel contents within this range were also associated with SFRC mixtures with higher fiber dosages when the fiber length was 40 mm;Low binder contents are generally associated with SFRC mixtures in which the sand and gravel contents are not too dissimilar, the coarse-to-fine aggregate ratio being in the range of 0.7–1.3. On the other hand, SFRC mixtures with moderate to high binder contents generally have significantly less gravel than sand;The superplasticizer content values in the database used for this study presented significant scatter, because it included some binder-intensive SFRC mixtures and, furthermore, there can be big differences between superplasticizers in terms of their recommended dosage range and optimal dosage. Despite this, the models developed worked well in estimating the superplasticizer content both in conventional SFRC mixtures and the more demanding ones.

## Figures and Tables

**Figure 1 materials-14-03900-f001:**
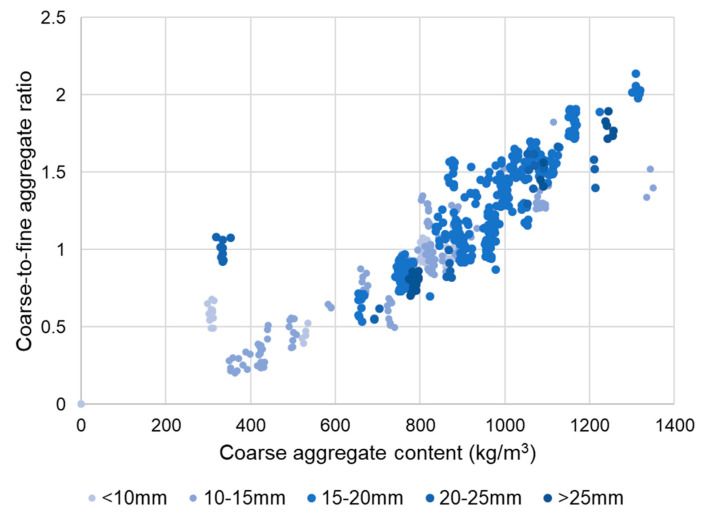
Coarse-to-fine aggregate ratio versus total aggregate content.

**Figure 2 materials-14-03900-f002:**
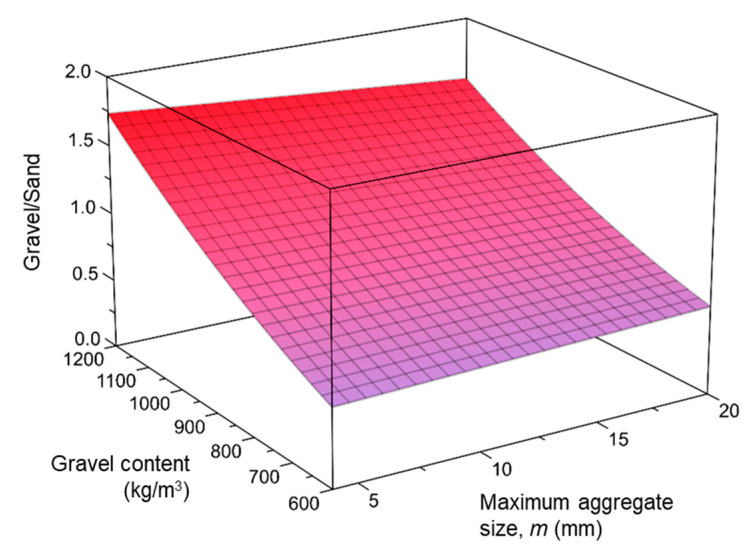
Coarse-to-fine aggregate ratio versus aggregate content and maximum aggregate size.

**Figure 3 materials-14-03900-f003:**
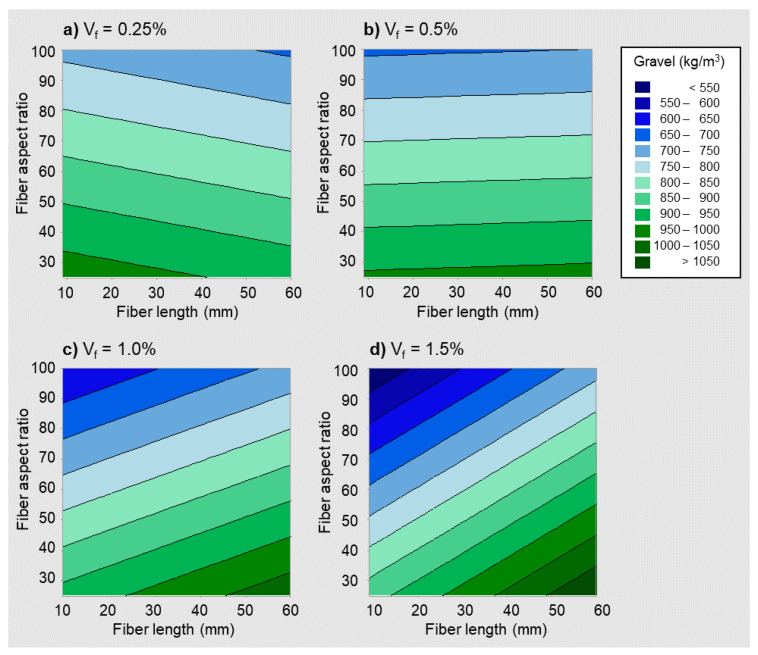
Coarse aggregate content versus fiber length and aspect ratio for different fiber dosages: contour plots for fiber volume fractions of 0.25% (**a**), 0.50% (**b**), 1.0% (**c**), 1.5% (**d**).

**Figure 4 materials-14-03900-f004:**
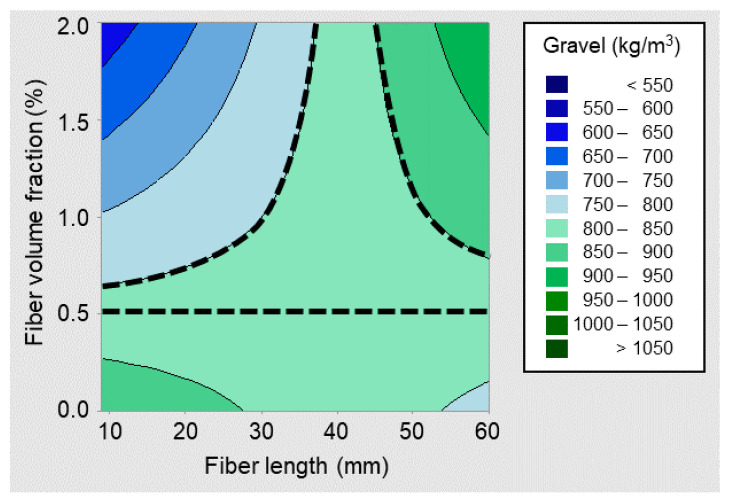
Coarse aggregate content against fiber length and volume fraction.

**Figure 5 materials-14-03900-f005:**
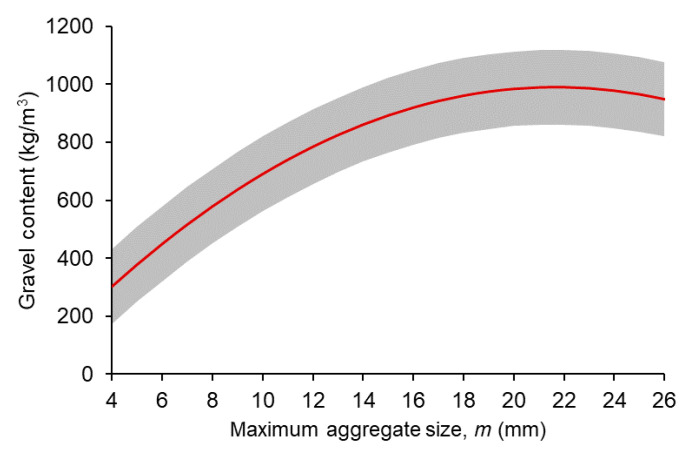
Coarse aggregate content versus maximum aggregate size.

**Figure 6 materials-14-03900-f006:**
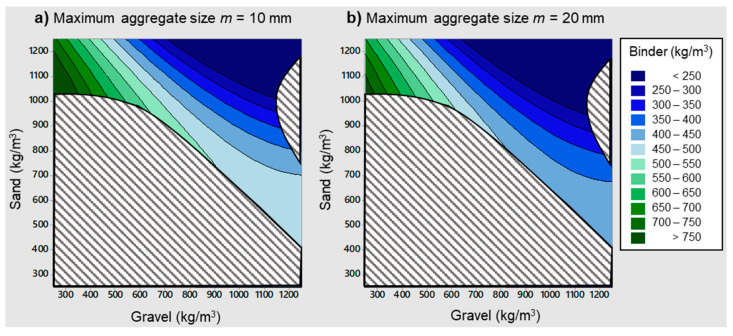
Binder content versus fine and coarse aggregate contents.

**Figure 7 materials-14-03900-f007:**
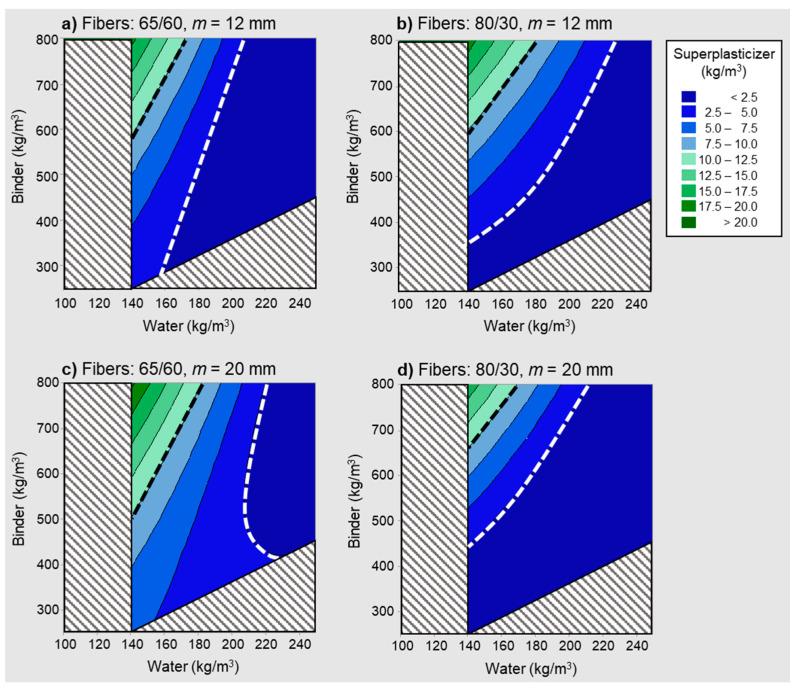
Superplasticizer content against water and binder contents.

**Figure 8 materials-14-03900-f008:**
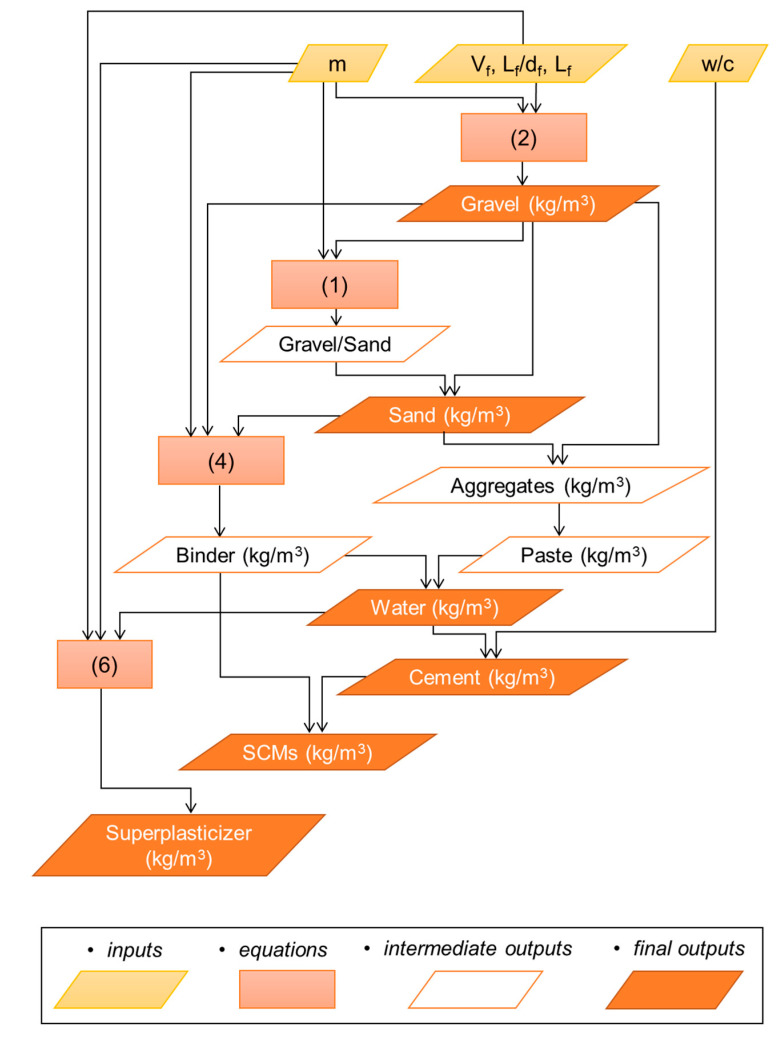
Flow chart of the proposed methodology for SFRC mix design.

**Figure 9 materials-14-03900-f009:**
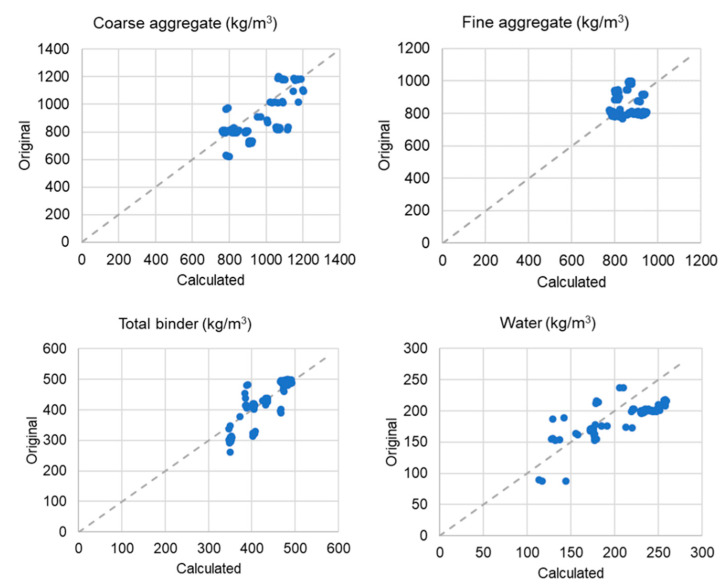
Evaluation of the model’s performance using a validation dataset.

**Table 1 materials-14-03900-t001:** Percentage of missing values for selected variables.

Variable	Missingness
Cement type	42.1%
Maximum aggregate size	23.9%
Superplasticizer dosage	4.8%
Fine aggregate content	3.2%
Coarse aggregate content	3.2%
SCMs content	2.6%
Cement content	2.6%

**Table 2 materials-14-03900-t002:** Representative values for the SFRC mixtures in the database.

Parameter	Median	Representative Range		Typical Range	
		Minimum	Maximum	Minimum	Maximum
Total binder (kg/m^3^)	440	270	710	390	550
Cement (kg/m^3^)	400	325	678	360	485
SCMs (kg/m^3^) ^1^	60	20	198	40	100
Water (kg/m^3^)	180	140	237	163	203
W/C ratio (non-dimensional)	0.45	0.22	0.60	0.36	0.50
Superplasticizer (kg/m^3^) ^2^	4.0	1.3	14.0	2.6	7.0
Fine aggregate (kg/m^3^)	835	524	1071	699	914
Coarse aggregate (kg/m^3^) ^3^	880	388	1157	774	999
Max. aggregate size (mm)	14	1	20	10	16
Fiber volume fraction (%)	0.51	0.25	2.0	0.43	1.0
Fiber length (mm)	45	13	60	30	60
Fiber aspect ratio (non-dimensional)	65	38	85	60	70

^1^ Values calculated after excluding mixtures which incorporated no SCMs. ^2^ Values calculated after excluding mixtures which had no superplasticizer. ^3^ Values based on mixtures which contained both fine and coarse aggregates.

## Data Availability

The complete list of sources from which the SFRC mixtures data were compiled is provided in the Appendix A that accompanies this paper. A copy of the spreadsheet with the database containing all these data is available on request from the corresponding author.

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
