# Peer review of "Meta-Analysis of Steel Fiber-Reinforced Concrete Mixtures Leads to Practical Mix Design Methodology"

_materials, 2021, doi:10.3390/ma14143900_

Round 1
Reviewer 1 Report
The authors conducted throughout research into a statistical model that helps determine the mixture design of SFRC based on metadata analysis of existing relevant publications. Detailed and in-depth literature review was provided. Proper methodology, which includes data-mining and statistical model development, was applied and explicitly presented. Ultimately, convincing results were obtained from the model, which was further strengthened with involved discussion. However, the authors should further clarify some of the issues discussed below before this manuscript being considered for publication.
My major concern about this manuscript is that the purpose of the proposed model is not clear, or robust. In the introduction section the authors seemed stress on the point that the proposed model is a “performance-based proportioning method”. However, no quantified performance of early-age or hardened-state of SFRC was taken into consideration, or at least explicitly discussed, in neither of data mining or model development process. There was some conclusion regarding performance in Section 4, such as “Increasing the fiber aspect ratio was found to be associated with decreasing the gravel content, which implied increasing the sand content and the relative volume of paste, that is, generally more cohesive mixtures” (Line 296-297), the performance outlook here, however, is obtained via empirical indirect induction not from model itself. Rather than providing any direct performance-related guidance, it read more like a tool that helps determine the mixture that is generally workable and with overall acceptable performance based on the range of the data domain, where no performance-related variables were input.
Some other minor questions:
Line 93-94. The authors mentioned that only publications that provided “sufficiently representative” are selected. Could the authors discuss briefly about what is the standard for “sufficiently representative”, is it performance-related?
As all the variables discussed affect each other statistically, the proportioning process discussed in Section 5 may not produce an optimal solution since the initial input, V_f, could be also affected by other variables that are determined later. Did the authors look into this? An iterative method (loop) may lead to more optimal solution.
Minor error/formatting issue:
Line 275. dddThe -> The?
Please unify the notation for maximum aggregate size: m in equations while M.A.S in all the figures.
Figure 8 needs to be adjusted, as it does not allow the standard scientific flow-chart format.
Reviewer 2 Report
In the Reviewer opinion the research paper entitled “Meta-Analysis of Steel Fiber-Reinforced Concrete Mixtures Leads to Practical Mix Design Methodology” is good.
The article presents the multiple linear regression applied to the statistical modeling of relationships between the mix design variables, leading to four equations that show considerable accuracy and robustness in estimating SFRC mixture proportions as a function of fiber content and dimensions, maximum aggregate size, and water-to-cement ratio. The main trends described by these equations are discussed in detail. The importance of the interactions between aggregates, supplementary cementitious materials, and fibers in proportioning SFRC mixtures as well as implications for workability and stability are emphasized. The predictive performance of these equations when used together as a data-driven mix design methodology is confirmed using a validation dataset.
Some comments which greatly enhance the understanding of the paper and its value are presented below. Specific issues that require further consideration are:
- The title of the manuscript is matched to its content but it is too long.
- The Introduction generally covers the cases.
- The methodology was clearly presented.
- In the Reviewer’s opinion, the current state of knowledge relating to the manuscript topic has been presented, but the author's contribution and novelty are not enough emphasized.
- Experimental program and results looks interesting and was clearly presented.
- In the Reviewer’s opinion, the bibliography, comprising 44 references, is rather representative.
- An analysis of the manuscript content and the References shows that the manuscript under review constitutes a summary of the Author(s) achievements in the field.
- In the Reviewer’s opinion the manuscript is well written, and it should be published in the journal after minor revision.
Reviewer 3 Report
This paper present Meta-Analysis of Steel Fiber-Reinforced Concrete Mixtures Leads to Practical Mix Design Methodology. Some modifications are mentioned here to enhance the quality of the manuscript:
1. I think the mechanical properties also need to be discussed in the paper. The significance of just analyzing the mix ratio is not great.
2. The hybrid fiber and micron fiber reinforced concrete are also important. So it is suggested to add some papers about this in the last two years in the introduction. For example[2, 1],
[1] Cao M, Si W, Xie C. Relationship of Rheology, Fiber Dispersion, and Strengths of Polyvinyl
Alcohol Fiber-Reinforced Cementitious Composites[J]. ACI Materials Journal. 2020, 117(3): 191-204.
[2] Li L, Li Z, Cao M, et al. Nanoindentation and Porosity Fractal Dimension of Calcium Carbonate Whisker Reinforced Cement Paste After Elevated Temperatures (up to 900℃)[J]. Fractals. 2021, 29(2): 2140001.
- What's the point of predicting the mix and how does that apply to engineering?
Round 2
Reviewer 3 Report
The paper has been sufficiently revised.